

# Multi-scale variability of southeast Australian wind resources

Claire Vincent[1,2], Andrew J. Dowdy[1,2]

[1]School of Geography, Earth and Atmospheric Sciences, The University of Melbourne, Melbourne, VIC, Australia
[2]ARC Centre of Excellence for Climate Extremes, The University of Melbourne, Melbourne, VIC, Australia

*Correspondence to*: Claire Vincent (claire.vincent@unimelb.edu.au)

**Abstract.** There is growing need to understand wind variability in various regions through the world, including in relation to wind energy resources. Here we examine wind variability in southeast Australia in relation to the El Niño-Southern Oscillation
(ENSO) as a dominant mode of atmospheric and oceanic variability for this region. The analysis covers variability from seasonal to diurnal time scales for both land and maritime regions of relevance to wind energy generation. Wind speeds were obtained from the 12 km grid-length BARRA reanalysis produced for the Australian region, with focus on wind at a typical hub-height of 100 m above the surface. Results show spatiotemporal variations in how ENSO influences wind speeds, including consistency in these variations over the wind speed distribution. For example, ENSO-related variations in mean
winds were mostly similar in sign to ENSO-related variations in weak winds, noting uncertainties for strong winds given available data. Diurnal variability in wind speed was larger for summer than winter and for land than ocean regions, with the diurnal cycle maxima typically occurring in the afternoon and evening rather than morning, plausibly associated with sensible heating of air above land following solar radiation. Localised variations in the diurnal cycle were identified around mountains and coastal regions. The results show some indication of ENSO influences on the diurnal variability. These findings are
intended to help enhance scientific understanding on wind variability including in relation to ENSO, as well as contribute information towards practical guidance in planning such as for use in energy sector applications.

## 1 Introduction

Wind variability is important to understand for many reasons, including given its key role in atmospheric circulation as well as noting the growth in electricity generation from wind turbines (Herbert et al. 2007; Blanco 2009; Manwell et al. 2010;
Sardorsky 2021). Large-scale atmospheric and oceanic modes of variability, such as the El Niño-Southern Oscillation (ENSO) and others, can influence weather conditions in regions through the world. For example, many studies have examined the influence of ENSO on regional temperature and rainfall variations (Ropelewski and Halpert, 1986; Bradley et al. 1987; Fraedrich, 1994; Davey, 2014; Taschetto et al. 2020). However, apart from trade wind variations in the tropical Pacific associated with the Walker Circulation that help define ENSO (Bjerknes 1969; Anderson et al. 2013; Chen and Wu 2013),
only a few studies have examined ENSO-wind relationships including with a focus on seasonal variations around Australia



(Dowdy 2020). Gunn et al (2023) used reanalysis data to examine the optimum distribution of wind farms in Australia to maximise night-time supply, and noted interannual variability in wind farm capacity factor of over 10%, which they attribute to ENSO, and Richardson et al (2023), who examined the co-variability of wind and solar droughts under different phases of ENSO, the Indian Ocean Dipole (IOD) and the Southern Annular Mode.


ENSO is a dominant global mode of atmospheric and oceanic variability on interannual time scales, with the El Niño phase of ENSO commonly referring to a weakening of the Walker Circulation and sea surface temperature gradients in the tropical Pacific, and conversely for the La Niña phase of ENSO, such as described by Holton and Dmowska (1989). Additionally, ENSO can influence winds in other regions away from the tropical Pacific region, including through teleconnections and

relationships between ENSO and other global and regional climate phenomena that influence local wind fields (Saji and Yamagata 2003).

El Niño conditions are typically associated with an intensification of the Hadley Cell, including constraining its meridional extension and an equatorial movement of the jet stream winds associated with its descending branch (Oort and Yienger, 1996;

Nguyen et al. 2013; Li et al. 2023). ENSO conditions are also related to variations in the Southern Annular Mode (SAM), characterized by a nearly zonally symmetric alternating pattern of pressure and geopotential height anomalies between high latitudes and midlatitudes in the Southern Hemisphere extratropics, associated with a meridional swing in the position of the eddy-driven westerly jet (Gong and Wang 1999; Lim et al. 2013). Another key climate mode that influences weather in the Australian region is the Indian Ocean Dipole (IOD), characterised by coupled atmospheric-ocean variations in the tropical

Indian Ocean (Saji et al. 1999). ENSO has significant relationships with IOD during the austral spring as well as with SAM during the austral spring and summer (e.g., Dowdy 2015).

In addition to large-scale atmospheric and oceanic mode of variability, wind variability can also be associated with a range of more localised processes including for the Australian region, such as katabatic (Manins and Sawford 1979; Low 1990) and sea

breezes (Masouleh et al. 2019). However, there is relatively little research to date on how ENSO might influence such localised wind features as those, including for the Australian region. Examples includes the study of Soderholm et al. 2017) who reported that El Niño (La Niña) tends to increase (suppress) the frequency of sea-breeze days in southeast Queensland and Rauniyar and Walsh (2024) showed a larger diurnal cycle of precipitation in El Niño than La Niña in tropical Australia Anomalies of mean sea-level pressure (MSLP) have also been examined in relation to ENSO for the Australian region during spring (Gillet

et al. 2023), with this also considered for different seasons in the analysis presented here given the potential to help provide insight on wind variations.

The relationship between ENSO and wind speed is examined here for southeast Australia, including the surrounding maritime region, noting a focus on offshore wind energy development in this region. Wind variability is analysed at seasonal time scales





and diurnal time scales, including how that variability may be linked to ENSO variability. This analysis is presented with a

focus on wind speed at a height of 100 m, given the relevance of this general height for wind turbines used in electricity

generation. Wind data are obtained from the BARRA reanalysis (Su et al. 2019), covering the Australian region based on a

horizontal grid spacing of ~12 km, as detailed in Section 2 on data and methods. The BARRA reanalysis was selected for use

here as it is specifically designed for providing insight on historical weather and climate for the Australian region. It is intended

that these findings based on the BARRA reanalysis could be complementary to studies that use other data sets, such as the

Wind Energy Atlas based on the CFSR reanalysis data with a horizontal grid spacing of about 40 km (Larsén et al. 2022).

This study is designed to help contribute to an enhanced understanding of wind variability, including in relation to ENSO as a

key driver of variability throughout the world and in the study region of focus around Australia. In particular, the data and

methods are selected to be relevant to wind turbines for power generation, including for insight on wind variability over both

land and offshore locations. In addition to energy sector applications, the study findings are also intended to help contribute to

broader guidance for planning and decision making in other sectors that can be affected by winds or associated ocean waves

and coastal impacts. This includes offshore shipping and other maritime activities, coastal recreation, ecology, fishing

aquaculture and natural resource management, in addition to helping contribute to an enhanced understanding of physical

processes and variations in the general circulation of the atmosphere.

## 2 Data and Methods

Homogenous wind data are not available at different heights throughout Australian land and maritime regions, with this study

using wind data obtained from a relatively fine-scale reanalysis dataset designed for the Australian region. The reanalysis

dataset used here is referred to as BARRA, as detailed in Su et al. (2019), based on model assimilation of local observations

data used in dynamical downscaling from ERA-Interim reanalysis data (Dee et al. 2005).

Wind speed data from the BARRA reanalysis are used here with a horizontal grid spacing of ~12 km in latitude and longitude.

The BARRA reanalysis has been found to improve on global reanalyses for many features, with verification studies showing

smaller root mean square errors (RMSE) in relation to observed surface pressure, temperature and wind speed (Su et al. 2019).

Cowin et al. (2023) compared the BARRA reanalysis to the ERA5 and MERRA-2 reanalyses at selected Australian Bureau of

Meteorology coastal weather stations, and found that BARRA had the lowest percentage error and highest correlation with

observations, while noting the difficulty in sourcing suitable observations for wind validation. Although some bias can still

occur in the reanalysis data, the study focus is on broad-scale climatological variability associated with ENSO, with other

aspects not in scope such as precisely quantifying wind magnitudes or long-term climate change trends. Extremely strong wind

speeds for short-duration gusts are also not considered, noting that type of analysis could be suitable for finer resolution



reanalysis data (e.g., ~4 km or finer grid spacing for convection-permitting simulations) that are not currently available for the Australian region.

Hourly time steps were used for the simulated wind data from BARRA reanalysis. Data from the lower 4 model levels are
used, corresponding to heights of 10.0 m, 36.7 m, 76.7 m and 130.0 m, with individual interpolation of wind speed from those heights to 100 m using a simple logarithmic scaling. This interpolation was done using a logarithmic wind profile assuming neutral conditions and a logarithmic constant of kappa = 0.4. After the interpolation of wind speeds for those four levels to 100 m, their average value was calculated for each time step at each grid cell. Those values were then used for the analysis presented here. This means that the wind data used here are indicative of a general height range from near the surface up to
about 130 m above the surface, noting relevance for infrastructure such as wind turbines for electricity generation that are typically built in this height range (Herbert et al. 2007; Blanco 2009; Manwell et al. 2010; Sardorsky 2021).

The diurnal variation (including amplitude and phase) of the wind speed is calculated here using a harmonic fit to the hourly data from BARRA reanalysis. This was done individually for each grid cell using the following steps. Firstly, the average wind
speed was calculated for each hour using all data through the study period 1990-2018. Secondly, a simple harmonic fit to the hourly values was applied, based on minimising the least squares difference to a sine wave with varying phase, amplitude and mean offset. Thirdly, the phase information is converted from Universal Time (UT) to a measure of local solar time (LST) using longitude in degrees east as follows: LST = UT + 24 * longitude/360. All diurnal cycles start at the model output time closest to 0000 LST, noting that can lead to a small discontinuity in some values at 150°E where the closest UT time to 0000
LST changes from 1500 UT (on the western side) to 1400 UT (on the eastern side).

ENSO is represented using monthly values of the NINO3.4 index, with data obtained from NOAA Physical Sciences Laboratory (https://psl.noaa.gov/data/correlation/nina34.anom.data; accessed November 2022). The study method is based on considering composite spatial fields for different phases of ENSO, including the El Niño and La Niña phases. While noting
there is no single globally accepted method to define these phases of ENSO, this study considers El Niño conditions to occur for NINO3.4 > 0.8 and La Niña conditions to occur for NINO3.4 < -0.8, with Neutral conditions for –0.8 ≤ NINO3.4 ≤ 0.8. The study analysis is presented using 3-month average vales to represent the austral seasons of summer for December, January and February (DJF), autumn for March, April and May (MAM), winter for June, July and August (JJA) and spring for September, October and November (SON). Table 1 presents the sample size of monthly values of NINO3.4 for each of the
ENSO phases (El Niño, La Niña or Neutral), presented individually for each of the four seasons, based on the study period 1990 to 2018.





Statistical significance of results is tested in this study using bootstrapping with 1000 randomised samples. Confidence in results is indicated in some figures with stippling to show values that are significantly different to what could be expected based on a random sample, using a statistical confidence level of 95% (two-tailed: i.e., > 97.5% or < 2.5%).

**Table 1: Sample size of monthly data for each of the ENSO phases considered here: El Niño, La Niña or Neutral. This is presented individually for each season (DJF, MAM, JJA and SON) based on the study period 1990 to 2018.**

| Season | El Niño | La Niña | Neutral |
|--------|---------|---------|---------|
| DJF | 17 | 24 | 46 |
| MAM | 8 | 10 | 69 |
| JJA | 7 | 7 | 73 |
| SON | 18 | 18 | 51 |

## 3 Results

### 3.1 Seasonal variability of wind speed and ENSO relationship

Figure 1 shows spatial fields of wind speed anomalies from the seasonal mean, based on composites for El Niño and La Niña phases of ENSO (noting the sample sizes for these phases from Table 1). Stippling is included to highlight regions that are significantly different to what could be expected on average based on a random sample (as detailed in Section 2). Results are presented individually for different seasons of the year, allowing for insight in cases where the ENSO influence might vary between different seasons in a given region.

The results indicate several regions with significant relationships between ENSO and wind speed for a given season. For the example of the austral summer (DJF), the winds tend to be weaker for El Niño and stronger for La Niña at about 39°S, particularly in some western maritime areas, with the converse anomalies to this occurring further poleward around 44°S. During winter (JJA), the winds are weaker for El Niño through much of the western and northern areas, with La Niña having stronger winds in the northeast and weaker winds in the south of the study region, while noting a relatively small sample size for El Niño and La Niña in JJA (from Table 1). Relatively weak winds also occur for both El Niño and La Niña along the eastern and southeastern parts of the Australian continent shown here, noting that this corresponds to areas of higher elevation along the Great Dividing Range. During spring (SON), relatively weak winds occur in many locations for El Niño and for La Niña, associated with stronger winds in general for the Neutral phase of ENSO. These results are consistent with those of Gunn (2023), who show the correlation of capacity factor with the Nino3.4 index for the whole year. Their results indicate more wind power along the central southern coastal areas of Australia and inland areas of SE Australia under La Niña conditions, and higher capacity factors around the topography in southeast Australia under El Niño conditions.




**Figure 1: Composite fields of wind speed anomalies from the seasonal mean, shown individually for Neutral (left column), El Niño (middle column) and La Niña (right column) during the austral summer (DJF: upper row), autumn (MAM: second row), winter (JJA: third row) and spring (SON: lower row). Stippling is included to highlight regions that are significantly different to what could be expected on average based on random chance alone, using a statistical confidence level of 95% (two-tailed). Sample sizes for each case are as shown previously in Table 1.**

## 3.2    High and low extreme wind speeds and ENSO variability



The relationship between ENSO and wind extremes is examined in this section, including for weak winds and strong winds. Results are examined here for wind speeds less than 5 m/s and for wind speeds greater than 25 m/s, providing examples of how ENSO influences might potentially vary over different parts of the wind speed spectrum. The lower threshold, for winds less than 5 m/s, is around the approximate range where the power generation from a wind turbine is typically dropping lower than its rated power, while noting that this power curve can vary between different turbines. This is relevant for the concept of

'wind drought' for turbines, where power generation might reduce substantially due to very weak winds. The upper threshold, for winds greater than 25 m/s, is sometimes used for indicating potentially dangerous and damaging winds (Brown et al. 2023). It is also noted that wind turbines typically have a cut-out speed for very strong wind speeds, which varies between turbines, where the blades will be pointed into the wind to reduce potential damages (e.g., excessive strain that might exceed the turbine design standards).


    The results for weak winds (i.e., less than 5 m/s) show various seasons and regions where significant anomalies occur for different ENSO phases (Fig. 2). During summer (DJF), weak winds tend to occur more frequently for El Niño and less frequently for La Niña at around 39°S, particularly in some western maritime areas. This is a similar region to where the mean winds were relatively weak for El Niño and strong for La Niña, as seen previously from Fig. 1. It therefore appears that ENSO

influences a range of the wind speed distribution in a similar way in this region during summer, with weaker winds for El Niño and stronger winds for La Niña on average, as shown from the mean wind speeds and well as for very weak wind speeds. Further poleward around 44°S, weak winds tend to occur less frequently for El Niño than La Niña, noting that this also indicates some similarities to results from Fig. 1.

During winter (JJA), weak winds occur more frequently for El Niño through much of the western and northern areas, with La Niña having weak winds occurring less frequently in the northeast and more frequently in the south of the study region (once again noting relatively small sample sizes for JJA from Table 1). Weak winds also occur more frequently for both El Niño and La Niña, as compared to the case for Neutral ENSO conditions, along the eastern and southeastern parts of the Australian continent shown here (i.e., corresponding to the Great Dividing Range). These regional features for weak winds in winter are

broadly similar to what was seen previously for mean winds (from Fig. 1), including weaker winds for El Niño through much of the west and north, La Niña having stronger winds in the northeast and weaker winds in the south, as well as El Niño and La Niña both having relatively weak winds along the Great Dividing Range.

    During spring (SON) for El Niño, weak winds occur more frequently in parts of the west and northeast (Fig. 2), similar to the

regions where the mean winds were weaker (from Fig. 1). During spring for La Niña, weak winds occur less frequently through much of the southwestern part of the study region and over the Australian continent, broadly similar to regions where the mean winds for La Niña were weaker than average during spring (from Fig. 1).



These results are mostly consistent with those of Richardson et al (2023), who indicated a greater likelihood of autumn wind droughts (defined as days when the daily average wind speed falls below 4.2 m s⁻¹) in La Niña than El Niño years over a

similar region of southeast Australia to that shown as having increased frequency of light winds in this study. They also show noisy or weak signals in spring and summer and a greater likelihood of wind droughts in El Niño than La Niña in winter. Our results show little difference in likelihood of light wind days between ENSO phases in winter, but we note several differences with the study of Richardson et al (2023), including using hourly rather than daily data, and using BARRA rather than ERA5 data, as well as a different study period.


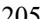

**Figure 2: The occurrence frequency of wind speeds less than 5 m/s, based on hourly time-steps. This is presented for Neutral (left column), El Niño (central column) and La Niña (right column), individually for the Australia summer (DJF: upper row), autumn (MAM: second row), winter (JJA: third row) and spring (SON: lower row). The values shown are the percentage difference in**

**frequency, relative to all times in the study period.**



The results for very strong winds (Fig. 3) are interpreted here with awareness of the limitations of model data for simulating some physical processes associated with storms and severe winds. In particular, the reanalysis data used here is not expected to accurately represent extreme winds from severe convective storms, as that could require much finer-scale model resolution. As such, these results are interpreted in relation to weather systems with larger spatiotemporal scales than thunderstorms, such as extratropical cyclones and frontal systems that can cause extreme winds in the region near southern Australia (Catto and Dowdy 2022). Additionally, results are shown in Fig. 3 only for locations where the average seasonal occurrence frequency of these extreme winds is 5 hours or more, so as to focus on results for larger sample sizes.

A substantial ENSO relationship with extreme winds is indicated in some maritime locations, including in the more southern parts of the study region for Neutral ENSO conditions during winter and spring. For El Niño, some southern regions with more frequent extreme winds as well as some with less frequent extreme winds, including depending on the season. The southern part of the study region is in the general latitude range where extratropical storms are relatively strong and frequent, such as described in Simmonds and Keay (2000). Wind speeds > 25 m/s are relatively rare, with many regions receiving < 10 hours per year above this level. The frequency of occurrence differences range between -0.4% and 0.4%, which amounts to only a few hours. Combining this analysis with convective-scale strong winds would lead to a greater number of hours with a wind speed > 25 m/s.

While acknowledging the limitations of the reanalysis data used here, such as discussed above and in Section 2 on data and methods, the results for extreme winds shown in Fig. 3 provide some indication that El Niño conditions could potentially be influencing extreme winds associated with the extratropical storm track in this region south of the Australian continent. This hypothesis provides scope for subsequent research that could be considered, such as using metrics for cyclone occurrence and intensity that might help examine these findings further.





**Figure 3: As for figure 2, but for wind speeds greater than 25 m/s. Only regions that have a frequency of greater than 10 hours per year are shown, since synoptic-scale winds of this magnitude are relative rare.**

### 3.3 Diurnal variability in wind speed and ENSO relationships

Figure 4 shows the amplitude of the diurnal variation in wind speed, presented as the average value for each season. The diurnal amplitude was calculated following the method described in Section 2 based on applying a harmonic fit to the average diurnal cycle in the BARRA reanalysis wind data. The harmonic fit of period 24 hours assumes a single daily peak in wind speed, and does not account for other periodicity such as semi-diurnal cycle or other higher-frequency components. Moreover, only some aspects of the diurnal cycle will be captured in this dataset. Vincent and Lane (2017) and Birch et al. (2016) have





shown errors in the timing and amplitude of the simulated diurnal cycle in of precipitation in the tropics across a range of model resolutions, and some of these errors will apply in the mid-latitudes as well. However, it is expected that the regions of afternoon, evening or nocturnal wind maxima will be reproduced, albeit with possible timing errors of up to several hours.

The results show much larger diurnal amplitude over land than the ocean in general. Over land, the amplitude tends to be
largest during the summer (DJF) and smallest during winter (JJA). This is also the case in some near-coastal ocean areas. Some land regions have larger diurnal variability than other regions, such as along the east and southeast of the continent (corresponding to the Great Dividing Range and eastern seaboard), in the island of Tasmania to the south of the continent, as well as in some northern and western locations. Relatively weak diurnal variability occurs in some inland regions shown here, as well as for ocean regions in general.


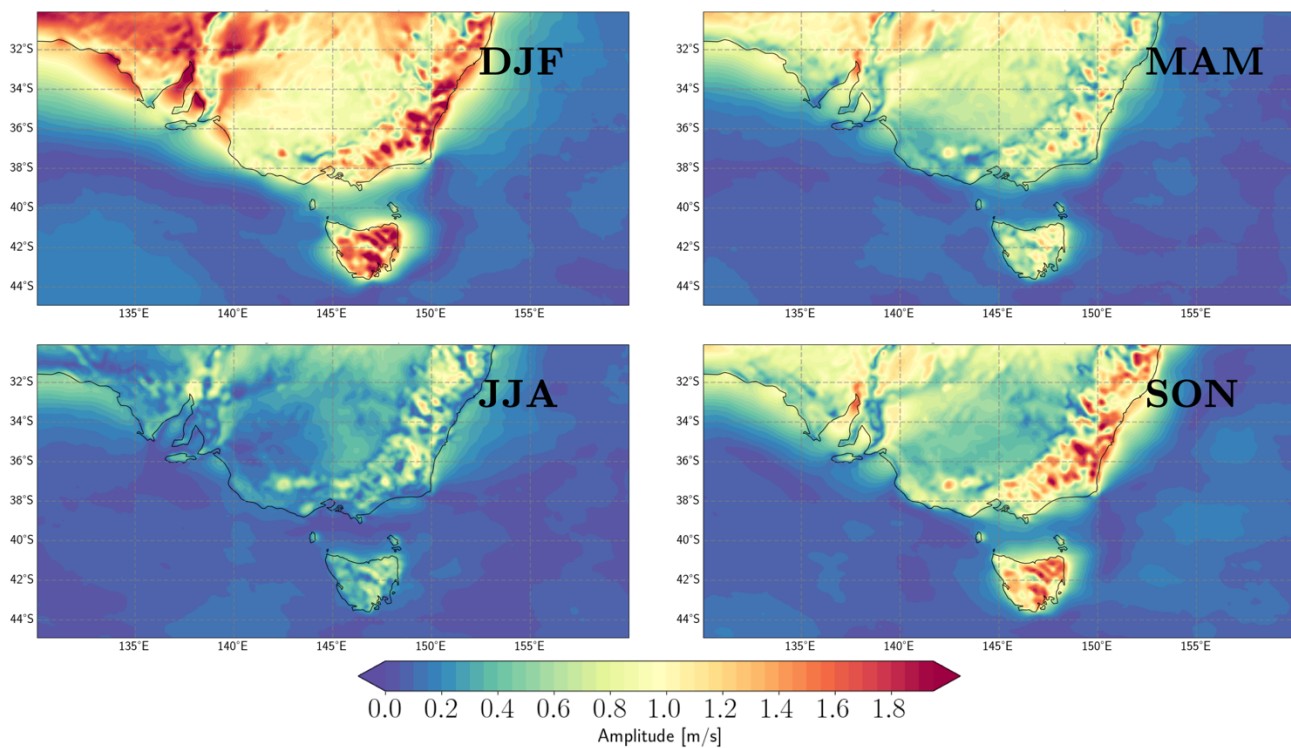

**Figure 4: Diurnal amplitude of wind speed, calculated from the harmonic fit to the average diurnal cycle at each grid point, presented as the average value for each season (DJF, MAM, JJA and SON).**

Figure 5 shows when the peak of the diurnal cycle in wind speed tends to occur during the day, presented as the average value for each season. This phase information uses the same harmonic fit to the wind data as used for the amplitude of the diurnal



variability (from Fig. 4). Results are presented in local solar time (LST) for a given longitude, rather than other options such as Universal Time, to allow a more consistent interpretation over this region.

The phase information for the diurnal cycle shows that peak wind speeds tend to occur in the afternoon and evening, rather than after midnight and in the morning. This includes early afternoon (e.g., from about noon to 18:00 LST) along the east and southeast of the continent (i.e., the Great Dividing Range and eastern seaboard region) and to the south for the island of Tasmania, as well as for some near-coastal regions in the west. Somewhat later timing for the peak winds occurs in regions further inland away from key orographic features such as coastal zones and the Great Dividing Range along the east coast of

the continent, with peak values occurring in the hours closer to midnight in general (e.g., from about 18:00 LST to 02:00 LST).

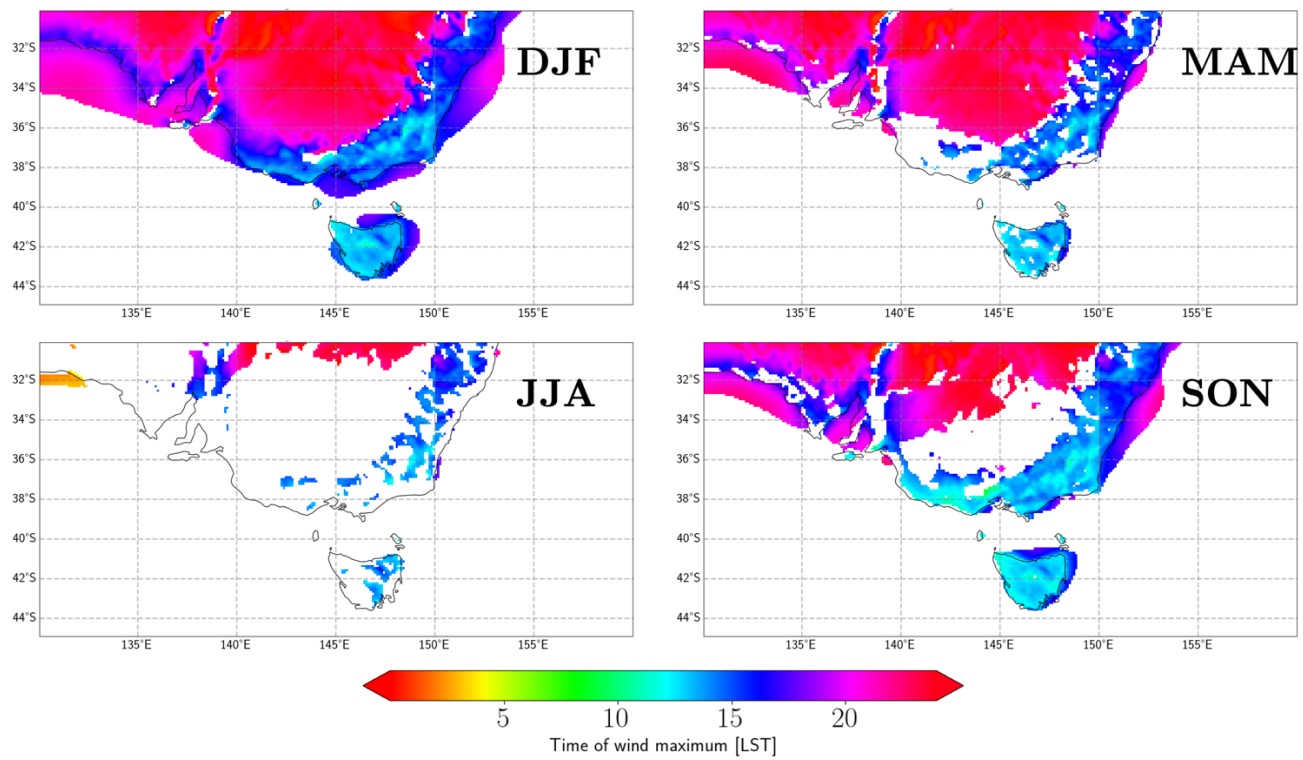

**Figure 5: Timing of the diurnal maximum in wind speed in local solar time (LST). This is presented as seasonal average values for DJF, MAM, JJA and SON. White regions represent locations where the amplitude of the diurnal variability is less than 0.5 m/s on**
**average for a given season (based on the amplitude values from Fig. 4).**




The influence of ENSO on the amplitude of the diurnal variation in wind speed is shown in Fig. 6. This is presented as seasonal average anomalies for each phase of ENSO, based on the same harmonic fits to the reanalysis wind data as used for Figures 4-5.


For DJF, the diurnal cycle is somewhat larger for El Niño and smaller for La Niña along the west and south of the continent, indicating opposing influences for those two ENSO phases. The largest anomalies for DJF occur for El Niño in some locations near the south coast of the continent. For MAM, positive anomalies for El Niño and negative anomalies for La Niña occur in the east of the continent and in some maritime regions to the west of the continent. For JJA, the anomalies for El Niño and La

Niña and similar in sign to each other in some regions, while noting a relatively small sample size for these cases (from Table 1), such as negative anomalies in some near-coastal regions and positive anomalies in some regions further inland in the continent. However, there is also some indication of La Niña having more negative anomalies than El Niño along the east and southeast of the Australia continent for JJA. Negative anomalies for La Niña in that region also occur during SON, with positive anomalies for El Niño.





**Figure 6: Anomalies of the diurnal wind amplitude, relative to all days, presented for El Niño (left panels) and La Niña (right panels). The values shown are seasonal averages for DJF (upper row), MAM (second row), JJA (third row) and SON (lower row).**




To help interpret some of the key features of the ENSO influence on diurnal wind speed (from Fig. 6), three individual regions are examined further here based on area-averaged wind speed for each hour of the day. This is presented in Figure 7, showing the average values for El Niño and La Niña conditions, and for each season, based on the monthly samples for each ENSO phase used previously as described in Table 1. This analysis has a focus on summer (DJF) including as this is when the largest diurnal variations occur (from Fig. 4) as well as noting summer is a key time of year for peak energy demand in Australia for

cooling as well as with natural hazards such as wildfires and severe wind gusts that can impact on supply (e.g., Dowdy et al. (2019)). Although selection of individual regions such as these is somewhat arbitrary, they were each selected here as contrasting examples based on having distinct diurnal cycle regimes, as indicated in Figures 4 and 5. with this analysis intended for indicative purposes to help provide some further insight on ENSO influences and diurnal wind variability. These regions also represent different orographic characteristics such as a near-coastal land region (Box 1), a region further inland (Box 2)

and a near-coastal maritime region (Box 3).

The diurnal variability for Box 1 in southeast (SE) Australia shows a peak around the middle of the day, shortly after noon (LST). There is relatively little variation between ENSO phases, apart from La Niña having a somewhat smaller diurnal range than the other ENSO phases (due to the weaker winds in the diurnal cycle being not as weak for La Niña compared to the other

ENSO phases). In contrast to Box 1, the winds in Box 2 for the inland Australia region peak closer to midnight. There is relatively little variation between ENSO phases, apart from La Niña having a somewhat smaller diurnal range than the other ENSO phases, in this case due to the stronger winds in the diurnal cycle being not as strong for La Niña compared to the other ENSO phases. For Box 3, the winds tend to peak in the late-afternoon, with El Niño having relatively weak winds throughout the day and La Niña having the weakest amplitude of the diurnal variability. The seasonal variability is much larger than the

inter-annual variability, with the largest diurnal cycle being found in DJF in all three boxes, reflecting the stronger solar forcing at this time of year that influences diurnal processes such as seabreezes, katabatic winds and nocturnal low-level jets. For example, Rife (2010) showed more low-level jet formation over Australia in DJF than JJA. Notably, at the inland box, the average DJF wind speed shows a lower minimum and a higher maximum that the JJA wind speed.



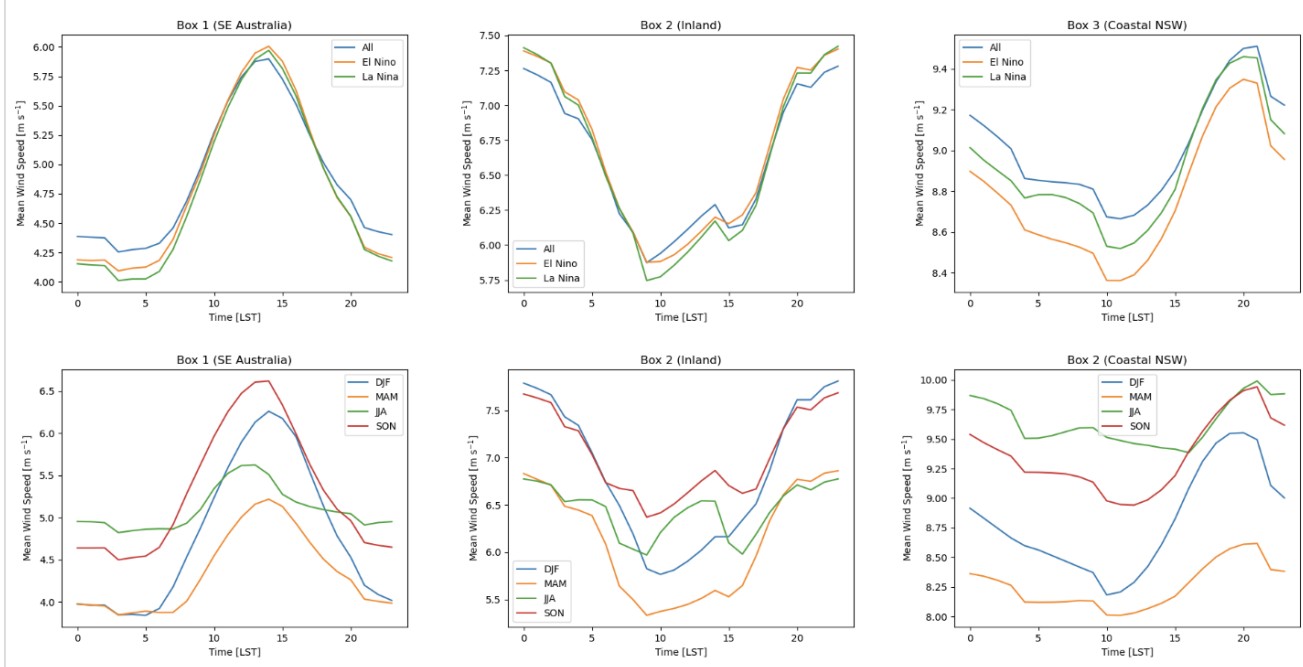

**Figure 7: Hourly wind speed values throughout the day, presented for area-average values in three regions (as shown in Fig. 8): Box 1 for southeast Australia (left panels); Box 2 in inland Australia (middle panels); and Box 3 for a maritime region near the central east coast of Australia (right panels). The upper panels show the diurnal cycle for summer conditionally averaged by ENSO phase, and the lower panels show the diurnal cycle conditionally averaged by season.**



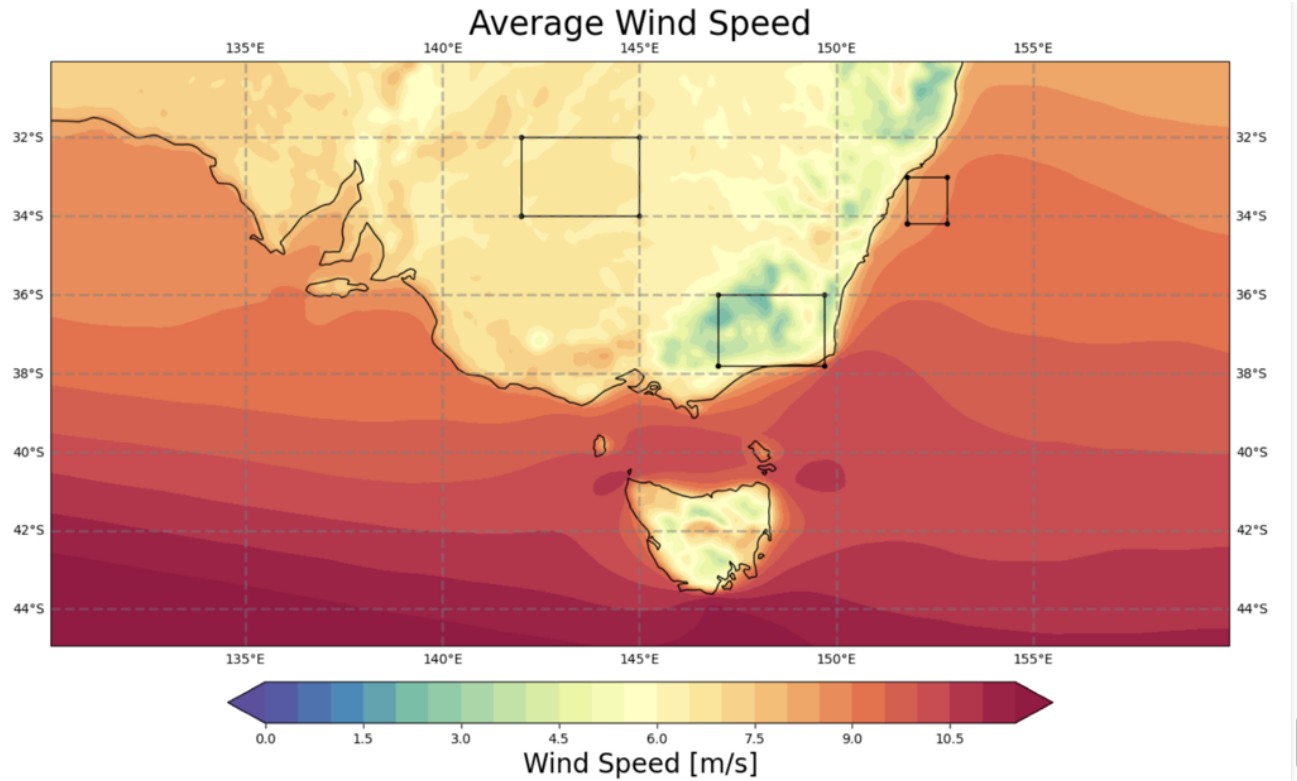

**Figure 8: Map showing the three regions of focus considered in Fig. 7, as well as mean wind speeds for the broader regional context. The three regions of focus are Box 1 in southeast Australia, Box 2 in inland Australia and Box 3 in a maritime location along the central east coast of Australia.**


### 3.4 Composite synoptic patterns

Figure 9 presents composite fields of mean sea-level pressure (MSLP) from the reanalysis data, shown individually for each season and ENSO phase to help provide some insight on physical processes associated with the ENSO-wind relationships. Similar to previous studies (Cai et al. 2011), this shows a clear variation through the year in the subtropical ridge, comprising

the latitude band with relatively high pressure, with low pressure systems typically on the poleward side in the extratropical cyclone storm track. This band of relatively high pressure is shown in Fig. 9 to move further poleward (i.e., to the south in Southern Hemisphere) during summer (DJF) as compared to winter (JJA) in general. For El Niño, the subtropical ridge during summer is shown to be somewhat stronger as well as further north compared to the case for La Niña, with a continuous ridge extending into western part of Bass Strait. This could plausibly be one contributing factor for somewhat weaker winds for El

Niño than La Niña in this region near southern Australia, similar to what was seen in Figures 1 and 2. Additionally in DJF, the pressure gradient in the far southwest of the region is stronger in general for El Nino than La Nina, noting this is also broadly consistent with the wind anomalies from Fig. 1.



The differences in pressure composites for different ENSO phases during MAM are not as notable as the case for DJF. This
suggests that relatively little wind variations might be expected to occur due to these relatively similar synoptic-scale conditions
between ENSO phases.

For winter (JJA), there is a stronger high-pressure region within the inland of the Australian continent for El Niño as compared
to other ENSO phases. This could plausibly be associated with more subsidence leading to relatively clear skies (i.e. less
cloudy conditions due to descending air), thereby causing a large diurnal range for temperatures and winds associated with
this. Additionally, the strong high-pressure system over the continent also might contribute to stronger pressure gradients and
associated wind speed to the south.

Spring tends to have relatively strong pressure gradients, including with a relatively zonal flow indicated here as compared to
other seasons. This is particularly the case during Neutral conditions in the southern parts of the study region. There is some
indication of higher pressure for El Niño than La Niña around the southern parts of the continent in spring, as well as relatively
low pressure in the southeast for La Niña, with these features being broadly similar to results presented in Gillet et al. (2023).

There are several other regions features apparent from Fig. 9, including notable details in geostrophic wind variations indicated
from these pressure composites. For example, in the region between the continent and the island of Tasmania to the south
(known as Bass Strait), the winds are predominantly westerly but can vary with a somewhat northerly or southerly component
depending on the ENSO phase and season. This poses questions around potential for orographic funnelling through this strait
and how that might vary in combination with ENSO influences.





**Figure 9: Composite fields of mean sea-level pressure (MSLP), shown individually for Neutral (left column), El Niño (middle column) and La Niña (right column) during the austral summer (DJF: upper row), autumn (MAM: second row), winter (JJA: third row) and spring (SON: lower row). Stippling is included to highlight regions that are significantly different to what could be expected on average based on random chance alone, using a statistical confidence level of 95% (two-tailed). Sample sizes for each case are as shown previously in Table 1.**



## 4 Summary and discussion

The variability of wind speeds was examined here using a regional reanalysis dataset designed for the Australia region. The study focus was on southeast Australia, including noting this as a key area of interest for wind energy generation. ENSO was found to have a range of influences on winds in this region around southeast Australia. This includes for different regions and
seasons based on mean winds as well as for extreme winds, with analysis presented for very weak winds as well as for very strong winds. ENSO was also found to influence the magnitude and the timing of the diurnal cycle of wind speed.

The findings showed that ENSO can influence a range of the wind speed distribution in a similar way in some cases. For example, during summer in the west of the study region (and around 39°S in general), winds tend to be weaker for El Niño
and stronger for La Niña, including for mean winds and for the occurrence frequency of very weak winds. MSLP anomalies indicated a stronger subtropical ridge for El Niño than La Niña around southern Australian during summer, as a potential contributing factor for these wind anomalies. For winter, weaker winds tend to be associated with El Niño in the west and north, with La Niña having stronger winds in the northeast and weaker winds in the south (e.g., around 44°S), as well as El Niño and La Niña both having relatively weak winds along the Great Dividing Range. During spring for El Niño, weak winds
occur more frequently in parts of the west and northeast, similar to the regions where the mean winds were weaker. During spring for La Niña, weak winds occur less frequently through much of the southwestern part of the study region and over the Australian continent, broadly similar to regions where the mean winds were relatively weak for La Niña during spring.

The study results highlight a range of influences on wind variability including from large-scale phenomena such as ENSO as
well as from smaller-scale aspects such as regional orographic features, including in relation to mean winds and the diurnal cycle. While some of the study results show a general zonal consistency, such as in the sign of the summer wind anomalies around 39°S for El Niño and around 44° S for La Niña (from Fig. 1), there are also clear spatial variations at any given latitude including around orography such as coastlines. Zonally-symmetric variations in this region can sometimes be related to the Southern Annular Mode (SAM), with a negative phase of SAM typically corresponding to a somewhat more equatorward
extent of features such as the Southern Hemisphere extratropical storm track region and the subtropical ridge as compared to the positive phase of SAM. As noted in the Introduction section, SAM variability can be related to ENSO variability, including with SAM tending to be in a negative phase for El Niño and in a positive phase for La Niña in the austral summer (e.g., Dowdy (2016)). The MSLP composite results presented in Fig. 9 showed a somewhat more equatorward latitude for features (e.g., such as the subtropical ridge position) for El Niño than La Niña. That direction of change (i.e., an equatorward shift in position)
is similar to what might be expected, given the tendency for negative SAM during El Niño conditions as noted above. This suggests a potential influence of ENSO on SAM as one contributing factor to pressure fields and resultant wind variations in this region. While acknowledging the limitations of the reanalysis data used here, particularly in relation to simulation of severe convective storms and associated extreme winds and boundary layer processes, the results for extreme winds shown in



Fig. 3 provide some indication that ENSO could plausibly be influencing extreme winds associated with the extratropical storm
track in this region south of the Australian continent (e.g., Simmonds and Keay (2000)).

For the diurnal cycle of wind speed, the results showed larger diurnal amplitude over land than the ocean in general, with larger amplitudes over land and in some near-coastal regions during summer than winter. Some land regions were found to have larger diurnal variability than other regions, such as along the east and southeast of the continent (corresponding to the
Great Dividing Range and eastern seaboard), in the island of Tasmania to the south of the continent, as well as in some northern and western locations.

The peak wind speeds tend to occur in the afternoon and evening, rather than after midnight and in the morning, with some spatial variation in this. For example, the peak winds occurred in early afternoon (e.g., from about noon to 18:00 LST) along
the east and southeast of the continent (i.e., the Great Dividing Range and eastern seaboard region) and to the south for the island of Tasmania, as well as for some near-coastal regions. However, the peak winds occurred somewhat later, closer to midnight, in regions further inland away from key orographic features such as coastal areas and the Great Dividing Range along eastern Australia.

These results provide some indication that solar radiation is having a substantial influence on the magnitude and timing of the diurnal cycle of wind speed. This includes noting the larger magnitude of diurnal cycle for summer and winter as well as for land than ocean regions in general. The timing of maxima for the diurnal cycle in the afternoon and evening (rather than morning) is also similar to expectations given the buildup of sensible heating of air above land during the afternoon following solar radiation through the day.


ENSO was found to have some relationship to the diurnal variability of winds in this region. The anomalies for El Niño and La Niña were mapped for individual seasons to detail the spatiotemporal features in how ENSO relates to the diurnal cycle of wind speeds throughout this study region.

The results presented here are intended to contribute towards a more detailed understanding of how ENSO relates to wind variability at different times through the year, including considering different parts of the wind speed distribution, as well as considering aspects such as the amplitude and phase of the diurnal cycle. Given the relative lack of previous studies that have focused on these aspects, the seasonal maps and analysis presented here could also be useful as part of guidance for wind energy planning. There is also considerable scope for future work that could build on these findings, such as around more
detailed understanding of physical processes and teleconnections associated with ENSO-wind relationships. Finer-scale reanalysis data could also potentially be considered or future studies, noting some plans underway for convection-permitting



reanalysis for the Australian region in coming years (Su et al. 2023), for more detailed analysis of extreme wind speeds from storms as well as other localised and fine-scale processes that could link wind variability with ENSO.

**Competing Interests**

The contact author has declared that neither of the authors has any competing interests.

**Acknowledgements**

This work was supported by the Australian Research Council Centre of Excellence for Climate Extremes (CLEX CE170100023) and the Zero Emissions Energy Laboratory of the Melbourne Energy Institute (MEI) at the University of
Melbourne. Analysis was conducted on the Australian National Computing Infrastructure (NCI). The authors would like to thank Holger Wolff of the CLEX computational modelling support team for assistance with the data processing.

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
