# Peer review of "Multi-scale variability of southeast Australian wind resources"

_EGUsphere, 2024_

## Author Comment (AC1)

**Responses to reviewer comments**

The authors thank the reviewers for providing these useful and insightful comments. Addressing these review comments has helped enhance various aspects in the revised manuscript. The responses to each review comment are detailed below on a point-by-point basis.

Reviewer RC1

The article compares predicted windspeeds for Southern EasternAustralia during El Niño and La Niña events, examining the frequency of both low windspeed and high windspeed events, based on data from the Barra dataset.

While the results of the study are interesting there are a few factors that would be worthwhile following up on:

1) The focus of the article is on windspeeds at a height of 100m, which is a little small for offshore wind turbines. It would be instructive to see how the results compare for a more typical offshore turbine hub height (150-200m)

**Authors' response: Thanks for this comment – this is a very good point. We performed additional analysis using wind speeds interpolated to 300m, which is similar to the height of planned offshore turbines. We found that the wind anomalies were very similar at 300m to those at 100m, although slightly stronger. We have mentioned this in the text (line 104-106), but not included the plots as they are quite similar to the 100 m results.**

2) While there is some mention of renewable droughts, the article does not discuss the effect on power output.

**Authors' response: The authors acknowledge that wind drought and power output more generally are important aspects to consider for energy systems. This study aimed to focus on wind variability, with results intended to have broad relevance for a range of different wind turbines that may have some variation in power output as a function of wind speed. The revised manuscript text now acknowledges this point raised by the reviewer, as follows (from lines 171-173 in revised manuscript in the first paragraph of Section 3.2),**

**"*This study's focus is on wind variability, with results intended to have broad relevance for a range of different wind turbines that may have some variation in power output as a function of wind speed. The lower threshold used here, for winds less than 5 m/s, is around the approximate range where the power generation from wind turbines is typically dropping lower than the rated power. This is relevant for the concept of 'wind drought' for turbines, where power generation might reduce substantially due to very weak winds, while acknowledging that analysis of power output is not a focus of this study.*"**

3) Likewise it would be instructive to understand how variability in windspeed (not just average speeds, and frequency of fast and slow events) change as a result of El Niño/La Niña.

**Authors' response: We agree. We have chosen to focus changes to the diurnal cycle under ENSO in this work, but there are other time-scales of interest – including hour-scale variability and synoptic-scale variability. We believe these are interesting and important topics, that warrant detailed studies in their own right. There is a broad range of potential metrics that could be selected to analyse variability in wind speed for different ENSO phases, including some that are shown in the manuscript (e.g., in Figures 7 and 8) as well as some that are not (e.g., metrics like standard deviation). Figure 7 shows the amplitude of the fit to the diurnal variations. As that metric relates to the magnitude of the variability, it was decided not to also include other metrics such as standard deviation that also relate to the magnitude of the variability. Additionally, the full shape of the diurnal cycle for three key regions is also presented in Figure 8, allowing the magnitude and the shape of the variability to be examined individually for different ENSO phases. The manuscript has been revised based on this review comment (see lines 299-305 when Fig. 7 is first mentioned in the text) to better communicate that Figures 7 and 8 are intended to help understand the variability in windspeed for different ENSO phases.**

4) It might be worthwhile indicating how diurnal maxima match with peak demand periods from the electricity network in figure 5 (now figure 6).

**Authors' response: This has been done with new text added to the revised manuscript. Lines 281-283 now state that:**

*"The afternoon and evening periods for peak wind speeds shown here are similar to peak periods for energy demand such as no hot summer days with air conditioner use particularly in afternoon and evening periods. **This timing is also distinct from the midday day peak in solar energy availability.** "*

Reviewer RC2
**General comments:**
This is a useful and clear study. The scope and discussion are appropriate, the results are novel, all the analyses and interpretation seem reasonable. It's a nice paper! I have provided quite a few specific and technical comments below but none of them take issue with the science, they are just to help improve the manuscript.

The only thing I feel uneasy about is how the choice of calculating the 100-m wind speeds from winds at other elevations impacts the results, especially for diurnal variability and in regions where the diurnal range in ABL stability is large (e.g., on land far from coasts). Maybe the authors are satisfied with the approach, but it isn't demonstrated in the manuscript and all the results depend on its veracity.

**Authors' response: The levels used for representing 100 m wind speeds are now more clearly explained in the manuscript, including noting that results for 300 m winds are also now included following some review comments above from RC1. In response to this comment and other related comments below, the method used for representing 100 m wind speed was refined as described in paragraph 3 of Section 2 Data and Methods, noting that this hasn't changed the findings in any substantial way. The model levels immediately below**

and above 100 m are used with logarithmic interpolation to 100 m, rather than using all lower model levels down to the surface, reducing the potential for boundary layer and near-surface influences on the analysis of 100 m wind speeds presented here. Similarly for the 300 m analysis, only the two model layers above and below a 300 m height are used with logarithmic interpolation to 300 m. Also see our reply and plot to the 'specific comment' for line 101.

**Specific comments:**

Line 32: In Gunn et al (2023) the interannual variability was not formally attributed to ENSO, but reasonably high absolute correlations were found with annual wind power and ENSO in parts of Australia.

**Authors' response: We have clarified this in the revised manuscript. Thank you for pointing this out. The revised text now reads**:

***"Gunn et al (2023) used reanalysis data to examine the optimum distribution of wind farms in Australia to maximise night-time supply, and noted interannual variability in wind farm capacity factor of over 10%, finding reasonably high absolute correlations between annual wind power and ENSO in parts of Australia".***

Line 45: This sentence about the ENSO-SAM linkage is a bit terse and maybe ambiguous. Maybe I am misinterpreting "zonally symmetric alternating", but it could be clearer to simply say what El Niño does to SAM and how it's expressed in pressure and geopotential height.

**Authors' response: The text was made more concise and clearer here by removing the word "alternating", as that was not essential to include.**

Line 101: I imagine a logarithmic Law of the Wall is fine with data averaged across a range of atmospheric stabilities temporally, but in detail each hourly wind profile may have very poor fits, especially at night, large absolute stabilities, near topographic obstacles, and near coasts. Of course, some choice must be made about how to interpolate the data for 100-m hub height, but it isn't demonstrated or explained why this is the best way for the problem at hand rather than an alternative (say, the simplest: linear interpolation between 76.7 m and 130 m). I think this Law of the Wall fit probably ends up dampening the temporal variability in the 100-m wind speeds compared to others (splines, differencing schemes, M-O theory, etc).

**Authors' response: As noted above, the model levels immediately above and below 100 m are used as described in the revised manuscript Section 2 Data and Methods, with this similarly being the case for analysis around 300 m (following other comments from RC1). There is also negligible difference in results when testing different interpolation method, including when interpolating between those two levels using a linear approach or using a logarithmic approach. This is shown in the figure below (not included in the paper), with differences less than 0.08 m/s throughout the study region. A logarithmic interpolation was selected for this study given this is not much harder to apply than a simple linear scaling, as well as because it may provide a somewhat more realistic representation of how the wind typically changes with height around these levels. We do acknowledge that this method may have larger errors in the presence of phenomena such as low level jets.  The text**

**describing our interpolation method and the sensitivity tests performed has been substantially rewritten (lines 101-109), as we realise that this was previously unclear.**

[Figure]

Figure R1: The magnitude of the differences between using a linear and logarithmic interpolation between adjacent heights for average wind speeds for a one year period.

Sections 3.1 & 3.2: It would be useful to provide up front what the absolute values for wind speed and frequency of <5 and >25 m/s winds are so that the anomalies can be contextualised. I recognise that average wind speed is given in Figure 8, and Figures 2 & 3 are normalised, but it would be useful for the reader if it was known a priori. Even a frequency distribution of wind speeds could be nice – this would help put the mean and tails in context, and verify the distribution looks Weibull.

**Authors' response: Thanks for this comment. We have now included a Weibull distribution for the modelled 100 m wind speeds over the land and water (Fig. 2 in the revised manuscript). This shows that the wind speeds > 25 m/s are in the extreme tail of the distribution. We decided to use to 22 m/s for the upper threshold instead, and have remade Fig. 3 (now Fig 4.) accordingly. Morever, an error in the masking of values where the frequency was < 10 hours per year was detected. This has been fixed in the new Fig 4. Consistent with the Weibull distribution, this almost never happens over land, which we attribute to the coarse model data that does not include convective-scale processes. Therefore, the differences are restricted to the sea.**

Figure 3: The 5 hours/season definition on Line 217 is inconsistent with the 10 hours/year in the caption. Masking out low frequency locations seems appropriate, but it should probably be done with a colour not on the colour map (e.g., grey, not white).

**Authors' response: This is now consistent in all places, for 10 hours/year. We have also shaded the masked regions in grey – this was a good suggestion.**

Figure 6: Is it possible to do the stippling and neutral conditions like Figures 1-3 here too?

**Authors' response: The neutral case has been added to this plot. We did not do stippling, because the amplitudes were based on a harmonic fit to the average diurnal cycle at each grid point. To do boot-strapping, we would have had to resample and then conduct multiple harmonic fits to the resampled data. This was quite computationally intensive, as it involves a separate fit for each gridpoint. Instead, we have shaded areas where the average diurnal cycle was < 0.5 m/s. This means that the ENSO differences in amplitude are only shown in areas where there is a large diurnal cycle.**

Line 338: I don't know if the "continuous ridge" apparent in Figure 9 during El Niño isn't apparent in La Niña simply because of which contour lines have been chosen – the shape (not magnitude) of the pattern looks very similar across both modes.

**Authors' response: The authors agree with the reviewer around this wording of "continuous ridge" not being accurate, with text removed here so that this is not discussed in the revised manuscript.**

Line 348: This is interesting but should be qualified by noting there's no stippling for this relationship.

**Authors' response: The lack of stippling and statistical confidence is now noted in the manuscript text at this point.**

Line 376: I don't think the results suggest that ENSO influences the timing of the diurnal wind speed cycle. I believe that only the top row of Figure 8 is presented for ENSO-dependent diurnal cycle: there's no phase shift there.

**Authors' response: This sentence was revised and now reads "*ENSO was also found to influence the magnitude of the diurnal cycle of wind speed, with no substantial influence apparent on the phase of the diurnal cycle*", based on results such as shown in the top row of Fig. 7.**

Line 413: Maybe this should be qualified by noting only regions where amplitudes are >0.5 m/s.

**Authors' response: This qualification is now noted in the text here.**

Line 422: How much is the timing related to low-level nocturnal jets? I would have guessed the flip from day to night peaks between boxes 1 and 2, respectively, was due to them being more prevalent over regions with lower relief and heat capacity. On the other hand, the amplitude of the relationship is lower in box 2. In any case, fitting a logarithmic profile to the data ensures this phenomenon isn't captured appropriately.

Authors' response: We agree that low-level jets could cause problems with the log-linear interpolation, and we have noted this in L108-109. However, we also note that the interpolation is between two levels that are quite close to 100 m (76m and 129m). A low level jet would likely influence multiple model levels within this part of the boundary layer, so we do not believe this effect would be particularly large.

**Technical corrections:**

Line 53: missing "s" on "mode".
**Authors' response: "s" added here.**

Line 56: parenthesis on reference missing, capitalised "Anomalies" (or maybe a full stop missing beforehand?).
**Authors' response: Full stop added.**

Line 89: this was only RMSE within Australia, yet the word "global" (for the other reanalyses coverage) make this ambiguous.
**Authors' response: It is now noted in the text here that the comparison with global reanalysis is only for the Australia region, given this is the region covered by the BARRA reanalysis (with BARRA produced based on dynamical downscaling from global reanalysis).**

Line 121: a reference for precedent of this definition of ENSO states would be useful.
**Authors' response: We chose a threshold of +/- 0.8 based on the approach used by the Australian Bureau of Meteorology for operational decision making and guidance provided to the public. We acknowledge that some other studies use different thresholds such as +/- 0.5. We tested the sensitivity of our results to different Nino3.4 thresholds. For smaller magnitude thresholds, the anomalies get larger, but statistical significance is lower due to smaller amounts of data.**
**http://www.bom.gov.au/climate/enso/indices.shtml**
**http://www.bom.gov.au/climate/ahead/about-ENSO-outlooks.shtml**

Line 150: "et al" missing for Gunn et al (2023) reference.
**Authors' response: "et al." was added for this case in the revised manuscript.**

Line 298: I think there's an erroneous "including" in this sentence.
**Authors' response: The word including was removed from this sentence.**

Line 318: I think the "that" should be a "than".
**Authors' response: Yes, thank you, this change was made.**

Figure 8: Could the boxes in the figure be labelled?
**Authors' response: This change was made.**

Line 341: The tildes on ENSO mode "n" characters are missing.
**Authors' response: This has been updated in the revised manuscript.**

Line 386: I think it's supposed to say "weak winds occur more frequently" for La Niña SON over the continent (as per Figure 2).
**Authors' response: This change was made.**